# Non-Invasive Cervical Spinal Stimulation and Respiratory Recovery After Spinal Cord Injury: A Randomized Controlled Trial with a Partial Crossover Design

**DOI:** 10.3390/brainsci15090982

**Published:** 2025-09-12

**Authors:** Hatice Kumru, Agustin Hernandez-Navarro, Sergiu Albu, Loreto García-Alén

**Affiliations:** 1Fundación Institut Guttmann, Hospital de Neurorehabilitació Institut Guttmann, Institut Universitari de Neurorrehabilitació Adscrit a la UAB, Camí Can Ruti S/N, 08916 Badalona, Spain; ahernandez@guttmann.com (A.H.-N.); salbu@guttmann.com (S.A.); lgarcia@guttmann.com (L.G.-A.); 2Universitat Autònoma de Barcelona, 08193 Barcelona, Spain; 3Fundació Institut d’Investigació en Ciències de la Salut Germans Trias i Pujol, 08916 Badalona, Spain

**Keywords:** transcutaneous spinal cord stimulation, respiratory function, cervical spinal cord injury

## Abstract

**Background/Objectives**: Respiratory impairment is the leading cause of morbidity and mortality in participants with spinal cord injury (SCI). Cervical SCI (cSCI) severely compromises respiratory function due to paralysis and weakness of the respiratory muscles. Recent evidence suggests that transcutaneous electrical spinal cord stimulation (tSCS) may enhance motor strength and promote functional recovery. Therefore, cervical tSCS, applied at cervical segments, holds potential as a therapeutic strategy to improve respiratory function in participants with cervical SCI. **Methods**: This randomized controlled trial with a partial crossover design included participants with both complete and incomplete cSCI. Neurological assessments were used, as well as tests to evaluate pulmonary function maximum inspiratory pressure (MIP), maximum expiratory pressure (MEP), and spirometric measurements. These assessments were conducted at baseline and after the last session. The experimental group received tSCS at the C3–C4 and C6–C7 cervical spinal levels, delivered at a frequency of 30 Hz during occupational therapy. The control group underwent identical occupational therapy sessions without stimulation. Each session lasted 30 min and was conducted over eight days. **Results**: Fifteen participants with cSCI received tSCS, while 11 cSCI participants were included in the control group. Seven participants took part in both groups. Only the tSCS group showed significant improvements in MIP, MEP, and forced vital capacity (*p* < 0.05), while no significant changes were observed in the control group. **Conclusions**: tSCS applied at the cervical segments can promote respiratory function following cervical SCI. This approach may support neuroplasticity and help reduce long-term respiratory complications in participants with cervical SCI. However, to confirm these effects, long-term stimulation protocols and follow-up studies in larger SCI populations are required.

## 1. Introduction

Respiratory dysfunction remains the leading cause of morbidity and mortality in participants with spinal cord injury (SCI) [1]. The respiratory neural network spans the cervical, thoracic, and lumbar spinal segments and receives descending input from brainstem respiratory centers [2,3]. The degree of respiratory impairment correlates strongly with both the neurological level and the completeness of the injury.

SCI is classified as either complete or incomplete, with the latter being more common. In incomplete injuries, portions of the respiratory spinal circuitry may remain intact, allowing for partial functional recovery. In contrast, complete injuries often lead to a total loss of sensorimotor function, including respiratory control, resulting in severe dysfunction [4,5]. Patients with injuries at or above the C5 level typically require ventilatory support due to diaphragmatic paralysis [6]. The diaphragm is the primary muscle of inspiration, while expiration is generally passive [7]. However, accessory muscles—including intercostals, abdominals, pectoralis major, scalenes, and sternocleidomastoids—play essential roles in supporting inspiration and maintaining ventilatory efficiency [7]. Participants with injuries below C5 may not require mechanical ventilation but often experience impaired respiratory function due to compromised accessory muscle activity. Common respiratory complications after SCI include reduced vital capacity, ineffective cough, decreased lung and chest wall compliance, and increased oxygen cost of breathing, all resulting from biomechanical and neuromuscular changes [4,5,8,9,10]. These issues contribute to long-term respiratory insufficiency and increase the risk of pulmonary edema, sleep-disordered breathing, and cardiovascular complications [11]. Therefore, effective respiratory management is crucial in both the acute and chronic phases of SCI.

Spinal cord stimulation (SCS)**,** originally developed for chronic pain, is now being explored for functional restoration in SCI. It delivers electrical impulses to modulate spinal neural networks [12]. Transcutaneous SCS (tSCS) is a non-invasive variant that delivers current through surface electrodes to activate spinal circuitry [13,14,15,16,17,18,19,20]. tSCS modulates spinal interneuronal activity, enhancing functional recovery in several neurological conditions [15,21,22,23,24,25,26,27,28,29,30,31]. Its mechanism is thought to involve activation of afferent fibers in the dorsal roots, increasing excitability of spinal networks without directly triggering action potentials [13,14,25,26,30]. Previous studies suggest that tSCS may improve respiratory outcomes in cervical SCI, although evidence remains limited to small cohorts and a single case report [27,32]. In one study, tSCS was delivered at cervical (C4) and thoracic (T9) levels combined with inspiratory training [27], while Gad et al. [32] applied tSCS at two cervical levels and T1.

Notably, in our previous study on upper-limb function with tSCS at two cervical levels [29], several participants with cervical SCI subjectively reported improved breathing, suggesting a potential respiratory benefit. Although these impressions were unpublished, they motivated us to include respiratory function in the current study. We hypothesize that cervical tSCS applied at two targeted spinal segments, even without respiratory training, may enhance respiratory function by facilitating activation of respiratory muscles in participants with cervical SCI. These improvements in the respiratory function may be associated with such factors as age, upper extremity or total motor score, time since injury, or tSCS intensity.

## 2. Materials and Methods

This study was a randomized controlled trial with a partial crossover design. Randomization was performed using a simple computer-generated procedure to ensure transparency, reproducibility, and minimization of allocation bias.

The participants were recruited in the functional rehabilitation program of the Institut Guttmann (Badalona, Spain). The inclusion criteria were as follows: (i) aged 18 years or older; (ii) diagnosed with a stable traumatic or non-traumatic cervical SCI; (iii) classified as AIS A, B, C, or D according to the American Spinal Injury Association (ASIA) Impairment Scale (AIS) [33]; (iv) time following SCI > 3 months, because stabilization of respiratory function occurs during the first 3 months [34] in patients who were under an active rehabilitation program; and (v) voluntarily participation in the research study.

The exclusion criteria were as follows: (i) a history of respiratory disorders prior to the SCI; (ii) mechanical ventilation; (iii) tracheostomy (iv) unstable SCI or other serious medical conditions (e.g., cancer, severe pulmonary or cardiac disease, etc.); (v) contraindications for tSCS; (vi) intolerance to electrical stimulation; (vii) peripheral nerve affectation and/or myopathy, which can impair respiratory function; and (viii) pregnancy.

Participants were recruited from the inpatient rehabilitation unit at the Guttmann Institute in Badalona, Spain, where the study was conducted. Ethical approval was obtained from the Guttmann Institute’s Ethics Committee, and all procedures followed the principles outlined in the Declaration (protocol code 2018279 and date of approval: 4 October 2019). This study was registered at ClinicalTrials.gov (Identifier: NCT07140354).

All participants with SCI were informed of all experimental procedures, after which each participants completed a signed informed consent process.

### 2.1. Neurological Assessment

The AIS scale was used to assess motor and sensory deficits based on the ASIA (American Spinal Injury Association) International Scale (AIS) [33]. We calculated the upper extremity motor score (UEMS, /50), as well as the total motor (/100) and total sensory score (/112). All assessments were conducted with participants in a supine position, in accordance with the International Standards for Neurological Classification of Spinal Cord Injury (ISNCSCI) [33].

### 2.2. Assessment of Respiratory Function

To evaluate respiratory function, we assessed maximal inspiratory pressure (MIP), maximal expiratory pressure (MEP), and spirometric parameters. MIP% and MEP% were calculated according to established reference standards and expressed as percentages of predicted values, adjusted for age, sex, and body size [35]. MIP is used to assess inspiratory muscle strength, while MEP evaluates expiratory muscle function, providing valuable insights into overall respiratory muscle performance. These pressures were measured using the MicroRPM Respiratory Pressure Meter (Micro Direct, Inc., Lewiston, ME, USA), and spirometric assessments were performed with the Datospir Micro spirometer (Sibelmed, Barcelona, Spain).

Spirometric measurements included forced vital capacity (FVC), forced expiratory volume in one second (FEV_1_), peak expiratory flow (PEF), and forced expiratory flow at 25%, 50%, and 75% of exhaled volume (FEF_25_%, FEF_50_%, and FEF_75_%, respectively).

FVC reflects the total volume of air forcibly exhaled after maximal inhalation. FEV_1_ measures the volume exhaled during the first second of forced expiration. FEF represents the airflow rate during the mid-phase of expiration, while PEF captures the maximum flow rate during a forced exhalation from full inspiration.

Respiratory function was assessed with participants seated comfortably, wearing nose clips, and with any restrictive clothing or belts loosened. After demonstrating the maneuvers, participants were instructed to inhale or exhale maximally for MIP and MEP recordings. Practice trials (2–3 trials) were allowed before final measurements were recorded. All evaluations were consistently conducted by the same person (LGA) to ensure methodological consistency.

Each participant underwent three recordings of MIP, MEP, and spirometric parameters at both baseline (pre-intervention) and after completing the intervention (post-intervention).

### 2.3. Experimental Design

The study was initially designed as a randomized, controlled clinical trial with two groups, namely (i) the intervention group, which received tSCS during occupational therapy, and (ii) the control group, which followed their routine occupational therapy program as part of standard rehabilitation without tSCS.

We used a computer-generated list for randomization, ensuring that the assignment of participants with SCI to treatment interventions was random. Participants in the control group could receive tSCS after their initial treatment following a minimum two-week washout. Two participants from the tSCS group were later included in the control group after a washout of at least two months and were treated as new participants, based on the literature indicating that tSCS effects plateau within two weeks [29] and show no further gains after one month [22].

Both interventions lasted two weeks, with participants attending four sessions per week, totaling eight sessions.

The study was conducted at the Guttmann Institute from October 2020 to April 2023. Clinical assessment of SCI, along with weight, height, and neurophysiological measurements, were performed only at baseline. All clinical and respiratory function assessments were conducted at baseline (pre-intervention) and after the final session (post-intervention).

### 2.4. Transcutaneous Electrical Spinal Cord Stimulation

tSCS was delivered using the BioStim-5 transcutaneous electrical stimulator (Cosyma Inc., Moscow, Russia), which generates biphasic, charge-balanced rectangular pulses (anodic-first, 1 ms per phase) with a 10 kHz carrier frequency, delivered at 30 Hz.

Stimulation intensity was set at 90% of the resting motor threshold (RT) of the abductor pollicis brevis (APB) muscle in the less affected hand, or in the right hand if both were similarly affected [29]. For single-pulse stimulation to record motor response of the APB, we used biphasic, charge-balanced pulses delivered anodic-first (positive phase preceding negative) with a 10 kHz carrier frequency. RT was defined as the minimum stimulation intensity required to elicit a spinal motor response with a peak-to-peak amplitude of ≥50 µV in at least 50% of 10 consecutive recordings, using standard electrodiagnostic equipment (Medelec Synergy, Oxford Instruments, Surrey, UK).

Stimulation was applied simultaneously at the C3–C4 and C6–C7 spinal levels using 2 cm diameter hydrogel adhesive electrodes (axion GmbH, Hamburg, Germany) as cathodes. Two 5 × 12 cm^2^ rectangular electrodes placed symmetrically over the iliac crests served as anodes.

tSCS was administered during occupational therapy in alternating intervals of 30 s of stimulation followed by 60 s of rest, for a total session duration of one hour.

Before each session, the stimulation intensity was gradually increased over several minutes to allow participants’ adaptation to tSCS.

Our routine occupational therapy included Armeo^®^Power (Hocoma, Volketswil, Switzerland) or hand training for 60 min per session [29].

In addition to hand training, the rehabilitation program included 4–5 h of daily therapy tailored to each participant’s needs. This comprehensive regimen comprised bipedal standing, gait training (with or without robotic assistance), hydrotherapy, trunk and core stabilization, upper and lower limb strengthening, stretching, occupational therapy, and balance and coordination training. Sessions were supervised by a multidisciplinary team to optimize functional recovery and support overall rehabilitation goals.

### 2.5. Data and Statistical Analysis

Data were collected for each participants following their respective assessments, and the analysis was performed after completing the final assessment. The means of MIP, MEP, and spirometric measures were calculated from three repetitions per participants. MIP% and MEP% were calculated as values expressed as percentages of predicted norms adjusted for age, sex, and body size [35]. The normal values were determined according to the reference values reported by Black and Hyatt [35].

Group data were expressed as mean ± standard deviation (SD), calculated separately for each group.

The normality of the data was assessed using the Shapiro–Wilk test. Variables with a normal distribution (UEMS, total motor score, and total sensory score) were analyzed using a 2 × 2 mixed ANOVA, with group (tSCS vs. control) as the between-subject factor and time (pre vs. post) as the within-subject factor. Post hoc comparisons were performed using paired Student’s t-tests to evaluate pre- versus post-intervention changes within each group.

For variables that were not normally distributed (MIP, MEP, MIP%, MEP%, and spirometric parameters within each group), the Wilcoxon signed-rank test was performed for paired comparisons to evaluate pre- versus post-intervention changes within each group, and 95% confidence intervals were calculated for MIP%, MEP%, and spirometric parameters. Effect sizes (Cohen’s *d*) were calculated for both neurological and respiratory assessments, with values interpreted as follows: 0.2 = small, 0.5 = medium, and 0.8 = large.

To assess score changes, we calculated the difference between pre- and post-intervention values. The Mann–Whitney U test was then used to compare these score changes between the tSCS and control groups. Spearman’s correlation analysis was performed between demographic and baseline variables—including age, UEMS, time since SCI, and tSCS intensity—and score changes in the tSCS group, including score changes in UEMS, total motor and sensory scores, MIP, MEP, MIP%, MEP%, and spirometric data. All statistical analyses were conducted using SPSS-16, with the significance level set at *p* < 0.05.

## 3. Results

A total of 19 out of 41 participants with cervical spinal cord injury (cSCI) were included in the study (Figure 1). Fifteen participants (fourteen men) received tSCS, while eleven participants (nine men) were assigned to the control group.

Five participants participated in both conditions, with at least a 1 month interval between them (range: 1–3 months), always starting with the control condition. In addition, two participants entered the control group after completing the tSCS intervention, following a washout period of at least 2 months (range: 2–4 months) (Table 1). For this reason, these participants were considered new participants.

All SCI participants completed the study without any dropouts. None of them reported abdominal muscle activation during stimulation, despite the anode electrode placement in the tSCS group.

The mean age was 37.2 years (SD: 13.5) in the tSCS group and 37.2 years (SD: 14.6) in the control group. The average time since injury was 5.7 months (SD: 2.2) in the tSCS group and 5.5 months (SD: 1.7) in the control group (Table 1).

Both groups were similar according to age, time since SCI, severity (AIS), and level of injury (Table 1).

### 3.1. Neurological Assessment

For UEMS, a 2 × 2 mixed ANOVA showed a significant main effect of time, F(1,24) = 21.22, *p* < 0.001, partial η^2^ = 0.469; both groups improved from pre- to post-intervention (tSCS 31.27→33.20; control 31.18→32.82). However, the main effect of group [F(1,24) = 0.002, *p* = 0.961] and the time × group interaction [F(1,24) = 0.15, *p* = 0.705, η^2^ = 0.006] were not significant. A paired Student’s *t*-test showed significant post-intervention improvements compared to baseline in both the tSCS group (*p* = 0.0008) and the control group for UEMS (*p* = 0.03) (Table 2, Figure 2).

For total motor score the 2 × 2 mixed ANOVA yielded a significant main effect of time in each group, F(1,24) = 9.63, *p* = 0.005, partial η^2^ = 0.286; means increased from 50.27→53.33 (tSCS) and 40.00→42.27 (control). However, the main effect of group [F(1,24) = 1.18, *p* = 0.287, η^2^ = 0.047] and the time×group interaction [F(1,24) = 0.21, *p* = 0.649, η^2^ = 0.009] were not significant. A paired Student’s *t*-test showed significant post-intervention improvements compared to baseline just in the tSCS group (*p* = 0.02) but not in the control group (*p* = 0.11) (Table 2, Figure 2).

For total sensory score, the 2 × 2 mixed ANOVA data were essentially unchanged (tSCS 71.93→72.27; control 59.30→58.70), with no significant effect of time [F(1,23) = 0.08, *p* = 0.784, partial η^2^ = 0.003], group effect [F(1,23) = 1.45, *p* = 0.242, partial η^2^ = 0.059], or time × group interaction [F(1,23) = 0.94, *p* = 0.342, partial η^2^ = 0.039] (Table 2, Figure 2). The paired Student’s *t*-test did not reveal any significant post-intervention changes in either group (tSCS: *p* = 0.63; control: *p* = 0.32).

A very large effect size was observed for UEMS in the tSCS group. In contrast, the effect sizes for UEMS in the control group and total motor scores in both groups were moderate. Sensory scores showed no measurable effect in either group.

### 3.2. Assessment of Respiratory Function

Baseline values of MIP, MEP, MIP%, MEP%, and spirometric measurements did not differ between the control and tSCS groups (*p* > 0.05, Mann–Whitney U test).

According to MIP%, at baseline in the tSCS group, 3 participants (20%) had normal inspiratory strength, while 12 (80%) showed pathological inspiratory strength. Following the last session, one additional participant achieved normalization of MIP%, resulting in a total of four participants who showed normal inspiratory strength (26.7%) [35].

For MEP% in tSCS, all 15 participants (100%) exhibited pathological expiratory strength [35], with no changes observed after the last session.

In the control group, two participants (20%; one female) had normal MIP%, while eight (80%) showed pathological values; these proportions remained unchanged after the last session. For expiratory strength (MEP%), all 10 participants with SCI (100%) exhibited pathological values [35], with no changes observed after the last session.

After the last session, MIP, MEP, MIP%, and MEP% improved significantly in the tSCS group (*p* < 0.05 Wilcoxon *t*-test), but not in the control group (*p* > 0.05 for each comparison Wilcoxon *t*-test) (Figure 3, Table 3).

tSCS led to an average improvement of 10.1% in MIP and 6.1% in MEP, whereas the control group showed minimal changes (0.8% in MIP and 0.5% in MEP). In the tSCS group, effect sizes were very large for MIP, MEP, and MEP%, while the improvement in MIP% was moderate. By contrast, no meaningful effect was observed in the control group.

Among the spirometric measurements, tSCS induced a significant increase in forced vital capacity (FVC) after eight sessions (*p* = 0.01) (Figure 4, Table 4a). However, no significant changes were observed in the other spirometric measurements. The control condition did not produce any changes in spirometric measurements (Table 4a). The 95% confidence interval of spirometric measurements given in Table 4b.

In the tSCS group, a moderate effect size was observed for FVC, while the remaining spirometric variables showed small or no changes in both groups.

### 3.3. Score Changes Between Both Group

Score changes demonstrated significant improvements in MIP, MEP, MIP%, and MEP% in the tSCS group compared with the control group (Mann–Whitney U test, *p* < 0.05 for each comparison) (Table 5). In contrast, score changes in UEMS, total motor and sensory scores, and spirometric measurements did not differ significantly between groups (Mann–Whitney U test, *p* > 0.05 for all comparisons, Table 5).

### 3.4. Correlation Analysis

No significant correlations were found between demographic or baseline variables—including age, UEMS, time since SCI, and tSCS intensity—and score changes in the tSCS group, including changes in MIP, MEP, MIP%, and MEP%, or spirometric data (*p* > 0.05). The only exceptions were a negative correlation between score changes in tSCS intensity at C3–4 and changes in MIP% (r = −0.51, *p* = 0.048), and a positive correlation with FEV1/FVC (r = 0.56, *p* = 0.028).

## 4. Discussion

This study aimed to evaluate whether tSCS applied at two cervical spinal segments could enhance respiratory function in participants with cervical SCI. The results demonstrated significant improvements in both inspiratory and expiratory muscle strength, as well as in respiratory capacity, measured by forced vital capacity, in the group receiving tSCS. Notably, the improvements in both inspiratory and expiratory muscle strength were significantly greater in the tSCS group compared to controls. These respiratory gains were not observed in the control group, despite both groups showing similar levels of upper-limb motor recovery. Furthermore, the score changes for respiratory function were not associated with upper extremity or total motor or sensory scores in either group. These findings suggest that targeted cervical tSCS may offer a specific therapeutic benefit for respiratory function independent of general motor improvements in participants with cervical SCI.

### 4.1. Transcutaneous Spinal Cord Stimulation for Respiratory Function in SCI

Two previous studies have reported improvements in respiratory function with cervical tSCS in participants with cervical SCI [27,32]. Kumru et al. [27] observed comparable increases in MEP, MIP, and pulmonary capacity when tSCS was applied at both the cervical (C4) and thoracic (T9) levels. Their study included eleven participants, incorporated a control group, and applied a five-day intervention period with structured respiratory exercises during tSCS sessions. The stimulation intensity was set at 90% of the threshold for the APB muscle, consistent with the current study. The anodes were placed bilaterally over the iliac crests.

In contrast, Gad et al. [32] presented a single-case study in which cervical tSCS improved breathing and coughing in a patient with chronic tetraplegia. Stimulation was applied to two cervical levels and one thoracic level (T1), with the anodes positioned bilaterally over the shoulders. The stimulation intensity was dependent on the participant and aimed to elicit the greatest functional respiratory response; it was delivered for 60 min per day, five days per week, over two weeks. Although promising, the single-case design highlights the need for larger-scale clinical validation.

In this study, one hour of daily tSCS applied at two cervical segments over eight days —without any respiratory training—resulted in a 10.1% increase in maximal inspiratory pressure and a 6.1% increase in maximal expiratory pressure. In contrast, the control group showed minimal improvements, with only a 0.8% increase in MIP and a 0.5% increase in MEP. These findings offer compelling support for the specificity of spinal neuromodulation in targeting and enhancing respiratory function.

The selective improvements observed in the tSCS group suggest that stimulation may strengthen chest wall musculature, thereby enhancing respiratory capacity. After SCI, the spinal cord retains the ability to relearn motor behaviors through activity-dependent processes [36,37,38]. The success of epidural electrical stimulation in restoring motor function after SCI is believed to be partly due to activity-dependent plasticity within spinal and supraspinal networks [39]. A similar mechanism is proposed for tSCS, involving non-invasive activation of spinal neuronal circuits, likely through the recruitment of large-to-medium afferent fibers in the dorsal roots. This recruitment increases spinal network excitability without directly eliciting action potentials [40,41]. By modulating the excitability of these interneuronal networks, tSCS may facilitate functional recovery [13,14,15,16,17,18,19,20,21,22,23,24,25,26,27,28,29,30,31].

Repeated tSCS sessions may initiate adaptive neuroplastic changes, facilitating functional reorganization within the respiratory network [42,43]. These activity-dependent processes can lead to learned responses that persist beyond the stimulation period and across multiple spinal segments—an effect observed with both transcutaneous and epidural spinal stimulation [44,45]. The underlying hypothesis is that tSCS elevates spinal sensorimotor networks into a more responsive state, thereby promoting voluntary motor control and ultimately leading to lasting functional gains. This mechanism is particularly relevant for respiratory network reorganization in SCI [32].

On the other hand, tSCS electrodes over the iliac crests may directly activate expiratory abdominal muscles, potentially shortening exhalation and influencing external respiration [46]. Although we did not observe overt abdominal contractions in our study, we cannot exclude a contribution from anode-induced activation at the iliac crests, which may improve respiratory function.

Correlation analyses revealed no significant associations between demographic or baseline variables or overall tSCS intensity and changes in respiratory outcomes within the tSCS group. This suggests that individual characteristics and baseline functional status did not significantly influence the respiratory improvements observed with tSCS. However, one notable exception emerged: stimulation intensity, specifically at the C3 level, showed a moderate negative correlation with changes in MIP%, and a positive correlation with changes in the FEV1/FVC ratio. These findings may indicate that higher stimulation intensity at C3 is associated with enhanced airway function (as reflected by improved FEV1/FVC) but might be inversely related to improvements in inspiratory muscle strength (as suggested by the decrease in MIP%). While these isolated correlations are statistically significant, their clinical relevance remains uncertain and should be interpreted with caution. Further studies are needed to explore whether differential stimulation at specific cervical levels modulates distinct aspects of respiratory function.

### 4.2. Respiratory Dysfunction Following Cervical SCI

While improvements in MIP and MEP were significant in the tSCS group, changes in other spirometric measures beyond FVC were less consistent. Nevertheless, the observed increases in respiratory pressures are clinically meaningful, as they reflect enhanced inspiratory and expiratory muscle strength, which may improve cough efficiency, airway clearance, and reduce the risk of respiratory complications in participants with cervical SCI. These findings warrant further investigation in longitudinal studies with larger patient cohorts.

At baseline, most participants of this study demonstrated impaired respiratory muscle strength. According to predicted values, pathological reductions in inspiratory strength (MIP%) were present in 80% of the tSCS and of the control group, while expiratory strength (MEP%) was pathological in all participants across both groups. These findings are consistent with previous reports showing that cervical and upper thoracic SCI impairs both inspiratory and expiratory muscles [5], resulting in reduced vital capacity, ineffective coughing, and decreased lung and chest wall compliance [4,5,8,9,10]. These deficits contribute to long-term respiratory insufficiency, with increased risk for pulmonary edema, sleep-disordered breathing, and cardiovascular complications [11]. Effective respiratory management across both acute and chronic stages is essential. Our findings suggest that cervical tSCS, alone or in combination with thoracic stimulation [27,32], may enhance coordinated activity of the diaphragm, trunk, and abdominal muscles. This may support neuroplasticity and mitigate long-term respiratory morbidity, although validation through larger studies and long-term protocols remains necessary.

### 4.3. Effect of Transcutaneous Spinal Cord Stimulation on Upper Extremity Motor Function

In our study, significant improvements in UEMS were observed in both the tSCS and control groups, with no significant differences in score changes between the groups. Similarly, García-Alén et al. [29] reported improvements in UEMS and GRASSP, although only GRASSP changes were significantly greater in the tSCS group.

By contrast, three studies applying tSCS in chronic SCI reported improvements in hand force and/or function [15,22,30], as well as motor and sensory abilities [30], although these findings were not obtained under controlled conditions [22,30]. In another study, grip strength increased approximately twofold without stimulation and threefold during stimulation [15].

The improvements observed in the tSCS group were expected; however, similar gains in the control groups of both our study and that of García-Alén et al. [29] may be explained by the fact that participants were within the first year post-SCI, a period when spontaneous recovery is still possible [47]. In contrast, other studies included participants more than one-year post-injury [15,22,30], a stage when recovery is typically limited [47,48].

### 4.4. Limitations of the Study

This study has several limitations related to its design. First, although the sample size is relatively small, it represents the largest cohort to date investigating the effects of tSCS on respiratory function in participants with SCI, and it is the first to examine the effects of two-segment cervical tSCS. Previous studies included only eleven participants and less time of stimulation at the cervical and thoracic segments [27] or were limited to a single case report [32]. Second, another limitation was the wide range of injury severity (AIS grades A–D) and time since SCI (3–10 months) within both groups. Although no significant relationship was found between score changes in respiratory function and time since SCI, both factors can influence the expected neurological recovery and, therefore, represent potential confounding variables. Third, the lack of blinding represents a limitation, as the perceptible intensity of the electrical stimulation made participant blinding unfeasible. Fourth, potential crossover contamination occurred, as 37% (7 participants) of participants switched groups. This unbalanced flow raises the possibility of order or learning effects, and, thus, the findings should be interpreted with caution. Fifth, the control group received standard occupational therapy alone, while the tSCS group received tSCS during occupational therapy. Therefore, we cannot fully exclude that differences between groups may partly reflect additional therapy time or attention. Future randomized controlled trials including sham stimulation are needed to strengthen internal validity. Lastly, the study did not include follow-up assessments, leaving the long-term sustainability of the observed respiratory improvements unknown. Future studies incorporating extended follow-up periods are needed to confirm the durability of these outcomes.

## 5. Conclusions

This study provides clinically relevant evidence that two-segment cervical tSCS can improve respiratory muscle strength and function in participants with cervical SCI. These results support the use of tSCS as a non-invasive neuromodulatory approach for respiratory rehabilitation. Combining tSCS with targeted respiratory training may yield greater functional benefits and reduce long-term respiratory complications. However, further validation through larger clinical trials and extended follow-up is essential to confirm these promising findings.

## Figures and Tables

**Figure 1 brainsci-15-00982-f001:**
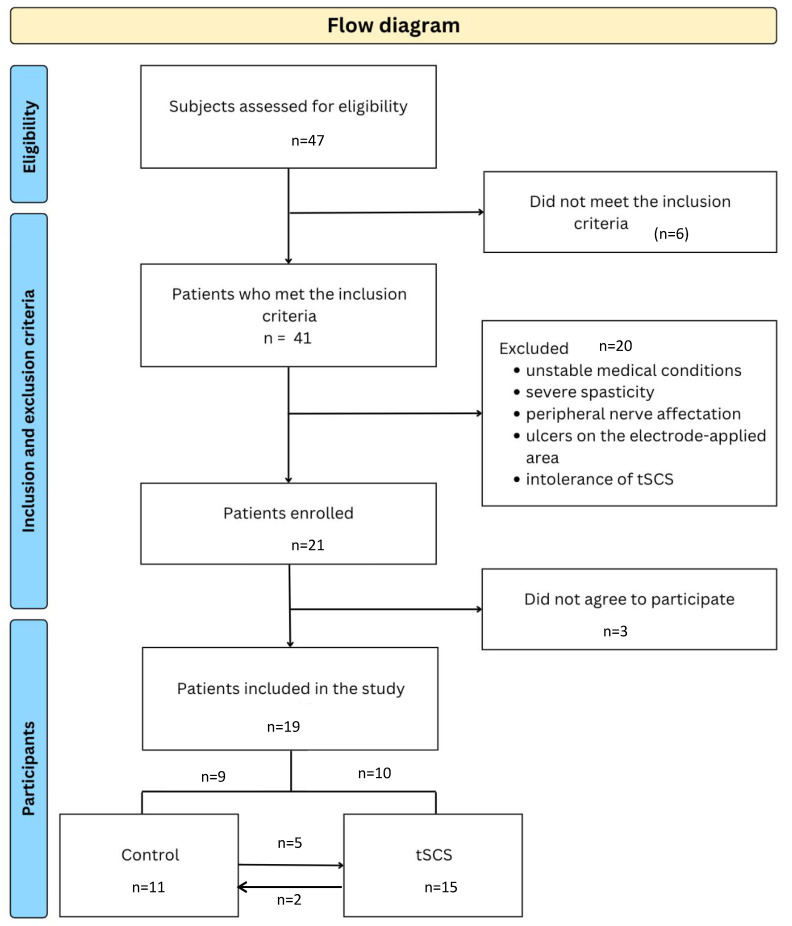
Flow diagram.

**Figure 2 brainsci-15-00982-f002:**
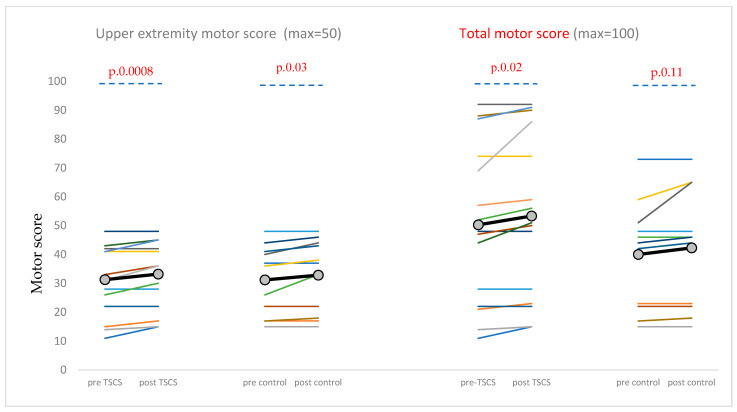
Changes in upper extremity motor score (UEMS) and total motor score. UEMS: upper extremity motor score. Individual data points are represented by the thin lines of different colors, while the group mean is indicated by a thick black line with gray circles. *p*-value was calculated using the paired Student’s *t*-test to compare pre- and post-intervention values in the tSCS and control groups.

**Figure 3 brainsci-15-00982-f003:**
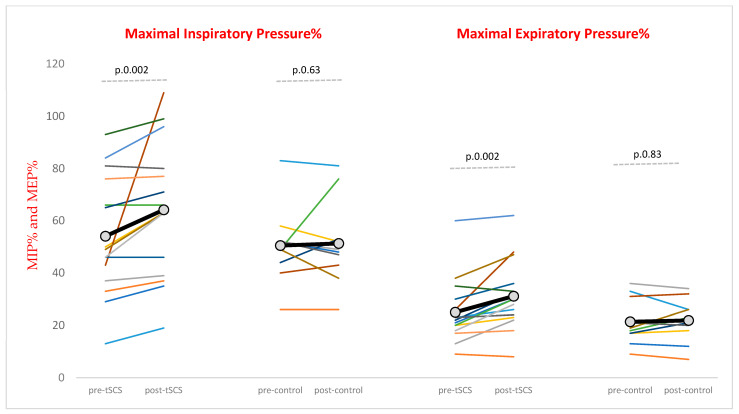
Maximal inspiratory pressure% and maximal expiratory pressure%. Maximal inspiratory pressure% (MIP%) and maximal expiratory pressure% (MEP%) represent values expressed as percentages of predicted value adjusted for age, sex, and body size. Data for each participant with SCI in the tSCS and control groups are shown at baseline (pre) and after the final session (post). Individual values are represented by the thin lines of different colors, while the group mean is indicated by a thick black line with gray circles.

**Figure 4 brainsci-15-00982-f004:**
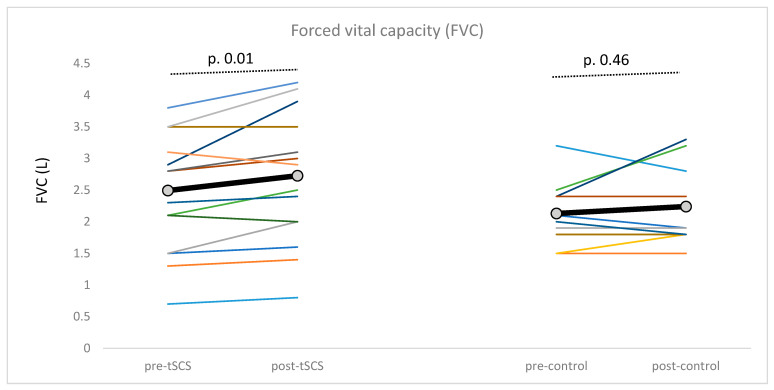
Changes in forced vital capacity (FVC) in tSCS and control groups. Individual data are represented by the thin lines of different colors, while the group mean is indicated by a thick black line with gray circles. *p*-value from the Wilcoxon *t*-test comparing pre- and post-last session measurements.

**Table 1 brainsci-15-00982-t001:** Clinical and demographic characteristics of participants with SCI, including the stimulation intensity used for tSCS.

	Age	Sex	Etiology	UEMS (/50)	NLI	AIS	Time Since SCI (Months)	tSCS Intensity	
								C3–4 Segment	C6–7 Segment
tSCS	25 **	M	T	11	C4	A	4	67	86
tSCS	46 **	M	T	15	C4	C	4	80	80
tSCS	36 *	M	T	14	C5	B	6	74	86
tSCS	36	M	T	41	C7	D	6	63	85
tSCS	18	M	T	28	C5	B	5	76	82
tSCS	28	M	T	26	C5	A	4	86	86
tSCS	28	M	T	48	C5	C	6	63	85
tSCS	38 *	M	T	33	C4	B	5	54	77
tSCS	56 *	M	No-T	42	C4	C	9	86	86
tSCS	60	M	T	43	C5	D	3	86	86
tSCS	21 *	M	T	22	C6	B	9	61	72
tSCS	22 *	F	T	43	C7	C	5	85	86
tSCS	55	M	T	41	C3	D	4	86	86
tSCS	47	M	T	31	C6	D	10	86	86
tSCS	42	M	T	31	C4	D	3	80	76
Mean	37.2			31.3			5.7	75.5	83.0
SD	13.5			11.8			2.2	11.2	4.6
Control	36 *	M	T	37	C4	B	5	-	-
Control	46 **	M	T	17	C4	C	6	-	-
Control	25 **	M	T	15	C4	A	8	-	-
Control	36	F	T	36	C4	C	6	-	-
Control	38 *	M	T	48	C7	B	3	-	-
Control	56 *	M	No-T	26	C4	C	6	-	-
Control	58	M	T	44	C6	A	6	-	-
Control	21 *	M	T	22	C6	B	8	-	-
Control	53	M	No-T	40	T1	B	3	-	-
Control	18	M	T	17	C6	B	4	-	-
Control	22 *	F	T	41	C7	C	4	-	-
Mean	37.2			31.2			5.4		
SD	14.6			12.1			1.7		
*p*	0.68			1.0			0.83		

M: male; F: female; T: traumatic; C = cervical (in the NLI column and for tSCS intensity column). AIS: American Spinal Injury Association (ASIA) Impairment Scale (AIS). NLI: neurological level of the injury, tSCS: transcutaneous spinal cord stimulation. SD: standard deviation; *: crossover—from control group to tSCS group; **: crossover—from tSCS group to control group. *p*-value for the comparison between the tSCS and control groups, calculated using an unpaired Student’s *t*-test.

**Table 2 brainsci-15-00982-t002:** Clinical changes in UEMS, total motor score, and sensory score.

Group	Age	UEMS (/50)		Total Motor Score (/100)		Total Sensory Score (/112)	
		Pre	Post	Pre	Post	Pre	Post
tSCS	25 **	11	15	11	15	24	20
tSCS	46 **	15	17	21	23	62	62
tSCS	36 *	14	15	14	15	92	92
tSCS	36	41	41	74	74	109	109
tSCS	18	28	28	28	28	22	22
tSCS	28	26	30	52	56	64	64
tSCS	28	48	48	48	48	67	67
tSCS	38 *	33	36	47	50	91	100
tSCS	56 *	42	42	92	92	112	112
tSCS	60	43	45	88	90	68	68
tSCS	21 *	22	22	22	22	75	75
tSCS	22 *	43	45	44	51	60	60
tSCS	55	41	45	87	91	73	73
tSCS	47	31	33	57	59	60	60
tSCS	42	31	36	69	86	100	100
Mean		31.3	33.2	50.3	53.3	71.9	72.3
SD		11.8	11.6	27.4	28.4	26.5	27.6
Effect size		1.10		0.70		0.13	
*p*		0.0008		0.02		0.63	
Control	36 *	37	37	73	73	62	62
Control	46 **	17	17	23	23	20	20
Control	25 **	15	15	15	15	71	71
Control	36	36	38	59	65	NA	NA
Control	38 *	48	48	48	48	67	67
Control	56 *	26	33	46	46	35	35
Control	58	44	46	44	46	26	26
Control	21 *	22	22	22	22	68	68
Control	53	40	44	51	65	112	106
Control	18	17	18	17	18	59	59
Control	22 *	41	43	42	44	73	73
Mean		31.2	32.8	40.0	41.4	59.3	58.7
SD		12.1	12.6	18.6	20.4	26.8	25.6
Effect size		0.74		0.53		−0.32	
*p*		0.03		0.11		0.34	

UEMS: upper extremity motor score. NA: not available. SD: standard deviation. *: crossover—from control group to intervention group; **: crossover—from intervention group to control group. Effect size: 0.2 = small, 0.5 = medium, and 0.8 = large. *p*-value was calculated using the paired Student’s *t*-test to compare pre- and post-intervention values in the tSCS and control groups.

**Table 3 brainsci-15-00982-t003:** Maximal inspiratory pressure (MIP) and maximal expiratory pressure (MEP) in the tSCS and control groups at baseline (pre) and after the last session (post).

**Age**	**Height**	**Weight**	**Group**	**MIP (cmH_2_O)**		**MEP (cmH_2_O)**		**MIP%**		**MEP%**	
	**(cm)**	**(kg)**		**Pre**	**Post**	**Pre**	**Post**	**Pre**	**Post**	**Pre**	**Post**
25 **	186	76	tSCS	37.3	45.3	52.0	72.3	29	35	21	30
46 **	174	76	tSCS	38.7	43.0	18.7	18.3	33	37	9	8
36 *	172	57	tSCS	45.7	47.7	29.7	33.7	37	39	13	22
36	175	95	tSCS	62.0	77.0	46.0	53.3	50	63	20	23
18	174	48	tSCS	17.0	24.7	57.3	66.0	13	19	23	26
28	176	71	tSCS	83.7	84.0	48.0	71.3	66	66	20	30
28	170	65	tSCS	82.7	90.7	53.3	77.3	65	71	22	32
38 *	165	54	tSCS	98.7	133.0	59.0	97.7	43	109	26	48
56 *	164	71	tSCS	91.3	100.0	48.7	51.3	81	80	23	24
60	180	77	tSCS	54.3	69.0	78.0	97.7	49	63	38	47
21 *	173	66	tSCS	60.0	60.7	73.3	89.3	46	46	30	36
22 *	170	51	tSCS	86.7	91.7	55.3	53.0	93	99	35	33
55	182	79	tSCS	94.7	108.3	126.7	130.3	84	96	60	62
47	172	82	tSCS	89.3	89.7	36.7	40.3	76	77	17	18
42	192	82	tSCS	55.3	75.3	41.3	63.7	46	63	18	28
			Mean	66.5	76.0	54.9	67.7	54.1	64.2	25.0	31.1
			SD	25.0	28.4	24.9	28.5	22.7	25.8	12.3	13.3
95% confidential interval								41.5–66.7	49.9–78.5	18.2–31.8	23.7–38.5
	effect size			1.05		1.09		0.62		1.00	
			*p*	0.001		0.001		0.002		0.002	
36 *	172	57	Control	61.0	55.7	29.0	28.7	52	48	13	12
46 **	174	72	Control	33.7	32.7	21.0	18.0	26	26	9	7
25 **	186	76	Control	67.7	63.7	86.7	81.3	52	49	36	34
36	167	61	Control	49.7	44.3	25.0	27.7	58	52	17	18
38 *	165	54	Control	101.0	98.7	75.7	59.0	83	81	33	26
56 *	164	71	Control	55.3	85.3	38.7	48.3	49	76	18	23
58	176	75	Control	49.0	59.3	35.0	42.7	44	53	17	21
21 *	173	66	Control	52.3	56.0	76.3	79.7	40	43	31	32
53	180	64	Control	58.7	53.3	45.3	43.0	52	47	21	20
18	179	61	Control	64.3	50.7	46.3	65.3	49	38	19	26
22 *	170	51	Control	NA	NA	NA	NA	NA	NA	NA	NA
			Mean	59.3	60.0	47.9	49.4	50.5	51.3	21.4	21.9
			SD	17.5	19.2	23.5	21.7	14.4	16.4	8.9	8.3
95% confidence interval								40.2–60.8	39.6–63.0	15.0–27.8	16.0–27.9
effect size				**0.06**		**0.13**		**0.07**		**0.13**	
			*p*	0.50		0.57		0.63		0.83	

tSCS: transcutaneous spinal cord stimulation. MIP: maximal inspiratory pressure. MEP: maximal expiratory pressure. NA: not available. SD: standard deviation. MIP% and MEP% represent values expressed as percentages of predicted value adjusted for age, sex, and body size. *p*-value according to a Wilcoxon *t*-test comparing pre- and post-last session measurements in each group. *: crossover—from control group to intervention group; **: crossover—from intervention group to control group. Effect size: 0.2 = small, 0.5 = medium, and 0.8 = large.

**Table 4 brainsci-15-00982-t004:** (**a**) Spirometric values at baseline (pre) and after the last session (post) in both groups. (**b**) 95% confidence interval (CI) of spirometric values at baseline (pre) and after the last session (post) in both groups.

**(a)**															
**Age**	**Group**	**Pre**	**Pre**	**Pre**	**Pre**	**Pre**	**Pre**	**Pre**	**POST**	**POST**	**POST**	**POST**	**POST**	**POST**	**POST**
		**FVC (L)**	**FEV1 (L)**	**FEV1/FVC (%)**	**PEF (L/s)**	**FEF50 (L/s)**	**FEF25/75% (L/s)**	**FEV1/FEV0.5**	**FVC (L)**	**FEV1 (L)**	**FEV1/FVC (%)**	**PEF (L/s)**	**FEF50 (L/s)**	**FEF25/75% (L/s)**	**FEV1/FEV0.5**
25 **	tSCS	1.5	1.4	92.0	1.4		2.2	1.4	1.6	1.5	89.8	3.7	2.4	2.2	1.2
46 **	tSCS	1.3	1.0	80.0	2.1	1.1	1.0	1.3	1.4	0.8	59.4	1.7	0.4	0.5	1.4
36 *	tSCS	1.5	1.1	69.7	1.7	1.0	0.8	1.5	2.0	1.4	70.4	2.0	1.3	1.2	1.7
36	tSCS	3.5	2.7	77.7	6.3	3.3	2.1	1.2	3.5	2.1	63.1	5.3	1.6	1.0	1.3
18	tSCS	0.7	0.6	93.1	1.8	1.3	1.1	1.1	0.8	0.7	95.2	1.6	1.1	1.1	1.2
28	tSCS	2.1	1.5	70.2	3.0	1.6	1.3	1.4	2.5	1.9	76.7	5.6	2.0	1.6	1.3
28	tSCS	2.9	2.8	92.2	5.4	4.0	3.6	1.3	3.9	3.4	86.8	5.7	4.0	3.7	1.4
38 *	tSCS	2.8	2.3	82.6	3.7	2.4	2.3	1.5	3.0	2.5	81.9	5.2	2.6	2.4	1.4
56 *	tSCS	2.8	2.1	76.0	3.7	2.2	1.8	1.4	3.1	2.4	79.1	5.0	2.7	2.2	1.3
60	tSCS	3.5	2.8	79.4	6.3	4.3	2.7	1.2	3.5	2.8	79.5	7.8	3.2	2.5	1.3
21 *	tSCS	2.3	2.0	89.3	3.6	2.7	2.5	1.4	2.4	2.1	88.2	3.9	2.5	2.3	1.4
22 *	tSCS	2.1	1.7	81.6	3.2	1.6	1.5	1.5	2.0	1.5	79.2	2.9	1.4	1.4	1.5
55	tSCS	3.8	1.3	33.2	7.4	3.2	2.5	1.5	4.2	1.9	45.6	2.1	1.8	1.7	1.8
47	tSCS	3.1	2.1	67.0	3.2	1.7	1.4	1.5	2.9	2.1	71.9	3.5	2.0	1.5	1.3
42	tSCS	3.5	1.7	46.7	2.2	1.0	1.0	1.7	4.1	2.7	66.9	4.5	2.0	2.0	1.6
	Mean	2.5	1.8	75.4	3.7	2.2	1.9	1.4	2.7	2.0	75.6	4.0	2.1	1.8	1.4
	SD	0.9	0.7	16.8	1.9	1.1	0.8	0.2	1.0	0.7	13.1	1.8	0.9	0.8	0.2
Effect size		0.61	0.24	0.10	0.23	0.14	0.06	0.01							
	*p*	0.01	0.86	0.88	0.15	0.62	0.88	0.73							
36 *	cont	2.1	1.8	88.4	3.6	2.1	2.0	1.4	1.9	1.6	85.7	3.5	2.0	2.0	1.3
46 **	cont	1.5	1.3	80.4	2.6	1.4	1.2	1.3	1.5	1.2	79.2	2.0	1.1	1.1	1.5
25 **	cont	1.9	1.8	94.8	4.6	3.2	3.1	1.2	1.9	1.8	92.1	4.2	2.8	2.6	1.3
36	cont	1.5	1.0	73.7	2.2	0.9	0.9	1.4	1.8	1.3	71.5	2.2	1.2	1.0	1.5
38 *	cont	3.2	2.5	78.2	4.7	2.4	2.2	1.4	2.8	2.3	82.6	3.7	2.4	2.3	1.5
56 *	cont	2.5	1.9	75.2	2.7	1.9	1.7	1.6	3.2	2.2	70.2	3.6	2.0	1.7	1.4
58	cont	2.4	1.7	71.8	3.7	1.4	1.3	1.4	3.3	2.6	77.8	4.5	2.9	2.4	1.5
21 *	cont	2.4	2.1	88.2	3.9	2.5	2.3	1.4	2.4	2.2	82.9	4.6	3.2	2.4	1.3
53	cont	NA	NA	NA	NA	NA	NA	NA	NA	NA	NA	NA	NA	NA	NA
18	cont	1.8	1.7	91.2	3.2	2.2	2.1	1.3	1.8	1.7	91.7	3.0	2.1	2.0	1.4
22 *	cont	2.0	1.8	89.7	3.4	2.0	2.0	1.4	1.8	1.7	91.2	3.2	2.2	2.1	1.3
	Mean	2.1	1.8	83.1	3.4	2.0	1.9	1.4	2.2	1.9	82.5	3.4	2.2	2.0	1.4
	SD	0.5	0.4	8.2	0.8	0.7	0.6	0.1	0.6	0.5	7.9	0.9	0.7	0.5	0.1
Effect size		0.25	0.30	−0.22	0.08	0.32	0.25	0.00							
	*p*	0.46	0.48	0.50	0.89	0.47	0.55	0.64							
**(b)**															
**Parameter**		**Pre-tSCS Mean ± SD (95% CI)**				**Post-tSCS Mean ± SD (95% CI)**			**Pre-Control Mean ± SD (95% CI)**			**Post-Control Mean ± SD (95% CI)**			
FVC (L)		2.5 ± 0.9 → [2.00, 2.998]				2.7 ± 1.0 → [2.15, 3.25]			2.1 ± 0.5 → [1.72, 2.48]			2.2 ± 0.6 → [1.74, 2.66]			
FEV1 (L)		1.8 ± 0.7 → [1.41, 2.19]				2.0 ± 0.7 → [1.61, 2.39]			1.8 ± 0.4 → [1.49, 2.11]			1.9 ± 0.5 → [1.52, 2.28]			
FEV1/FVC (%)		75.4 ± 16.8 → [66.1, 84.7]				75.6 ± 13.1 → [68.4, 82.9]			83.1 ± 8.2 → [76.8, 89.4]			82.5 ± 7.9 → [76.4, 88.6]			
PEF (L/s)		3.7 ± 1.9→ [2.65, 4.75]				4.0 ± 1.8 → [3.00, 5.00]			3.4 ± 0.8 → [2.78, 4.02]			3.4 ± 0.9 → [2.71, 4.09]			
FEF50 (L/s)		2.2 ± 1.1→ [1.59, 2.81]				2.1 ± 0.9 → [1.60, 2.60]			2.0 ± 0.7 → [1.46, 2.54]			2.2 ± 0.7 → [1.66, 2.74]			
FEF25/75 (L/s)		1.9 ± 0.8 → [1.46, 2.34]				1.8 ± 0.8 → [1.36, 2.24]			1.9 ± 0.6 → [1.44, 2.36]			2.0 ± 0.5→ [1.61, 2.39]			
FEV1/FEV0.5		1.4 ± 0.2 → [1.29, 1.51]				1.4 ± 0.21 → [1.29, 1.51]			1.4 ± 0.1 → [1.32, 1.48]			1.4 ± 0.1 → [1.32, 1.48]			

Cont: control group; tSCS: transcutaneous spinal cord stimulation; FVC: forced vital capacity; FEV_1_: forced expiratory volume in one second; PEF: peak expiratory flow; FEF: forced expiratory flow at 25%, 50%, and 75% of exhaled volume (FEF_50_% and FEF_25/75%_, respectively). SD: Standard deviation. CI: confidence interval. *p*-value from the Wilcoxon *t*-test comparing pre- and post-last session measurements. *: crossover—from control group to intervention group; **: crossover—from intervention group to control group. Effect size: 0.2 = small, 0.5 = medium, and 0.8 = large.

**Table 5 brainsci-15-00982-t005:** Score changes in clinical and respiratory assessments in control and tSCS group.

Score Changes			
	Control Mean (SD)	tSCS Mean (SD)	*p*
UEMS	1.6 (2.2)	1.9 (1.8)	0.50
Total motor score	2.3 (4.3)	3.1 (4.4)	0.30
Total sensory score	−0.6 (1.9)	0.3 (2.6)	0.68
MIP (cmH_2_O)	0.7 (12.1)	9.5 (9.1)	0.008
MIP%	0.8 (10.7)	10.1 (16.4)	0.01
MEP (cmH_2_O)	0.8 (9.6)	10.1 (16.4)	0.01
MEP%	0.5 (4.1)	6.1 (6.1)	0.01
FVC (L)	0.1 (0.4)	0.2 (0.3)	0.17
FEV1 (L)	0.1 (0.3)	0.2 (0.4)	0.36
FEV1/FVC (%)	−0.6 (3.3)	0.0 (8.9)	0.46
PEF (L/s)	0.0 (0.5)	0.3 (1.7)	0.21
FEF50 (L/s)	0.2 (0.5)	0.0 (0.9)	0.68
FEF25/75% (L/s)	0.1 (0.4)	0.0 (0.5)	0.76
FEV1/FEV0.5	0.0 (0.1)	0.0 (0.1)	0.75

SD: standard deviation. *p*-value was obtained from the Mann–Whitney U test comparing the score changes between the control and tSCS groups.

## Data Availability

All data generated or analyzed during this study are available in the following Figshare repository (7 August 2025): https://doi.org/10.6084/m9.figshare.29850395.v1.

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
