# Peer review of "Non-Invasive Cervical Spinal Stimulation and Respiratory Recovery After Spinal Cord Injury: A Randomized Controlled Trial with a Partial Crossover Design"

_brainsci, 2025, doi:10.3390/brainsci15090982_

Round 1
Reviewer 1 Report
Comments and Suggestions for Authors
Respiratory failure is a leading cause of death after spinal cord injury. Even small improvements can reduce pneumonia risk, hospitalizations, and mortality; hence, this is a very important field needing more research. Cervical transcutaneous spinal cord stimulation (tSCS) to improve respiratory function is a novel and emerging intervention gaining strong theoretical and early clinical support.
The novelty behind this research study is the application of a dual-site cervical tSCS protocol; comparing tSCS during occupational therapy (OT) with a control group undergoing OT alone. The study tracked changes in MIP, MEP, FVC, after eight 1-hr sessions over 2 weeks. Significant improvements in these respiratory measures were seen in the tSCS group. Other spirometric measures did not show any significant change for either group.
Major
Materials and Methods, pg 6, line 192. The current analysis does not clearly describe how the mixed design (partial crossover and parallel components) was handled. Since the study includes both paired (within-subject) and unpaired (between-group) participant types, it’s important to specify which tests were used for which analyses. Including crossover participants in both within- and between-group analyses can violate assumptions of statistical independence and risk Type 1 errors. For example, consider only using the first phase data from crossover participants in the Mann–Whitney U test.
While acknowledgment in the limitations section demonstrates awareness, the statistical analysis section does not clearly describe how this crossover data was managed to maintain statistical independence for the between-group comparisons. Please clarify and/or consider redoing the statistical analysis to improve robustness of the data presented.
Minor
Title. Considering the partial and unplanned crossover design, I feel that the manuscript title is a little misleading, stating that the study is an RCT with "high crossover rate". Please revise to something along the lines of ‘partial crossover design’ or ‘mixed design’.
The number of participants who crossed over doesn’t constitute as a “high” rate. See my comments for the methods section (pg 4, line 91).
Introduction, pg 2, line 80. You introduce one of your prior studies, Kumru et al 2023. To follow this as a foundation, it would be nice to read a description of how this current study is a step up. Please highlight the rationale/novelty of this current study.
Introduction, pg 3, line 85. Please cite prior studies your group performed with cervical SCI, reporting improvements in breathing. You only noted one, but the sentence reads as plural.
Materials and Methods, pg 4, line 91. 7 people is not a large absolute number to call the crossover “high”, especially in the absence of a pre-specified crossover design. 7 out of 19 enrolled is ~37%.
Furthermore, since randomization into TSCS and control groups only occurred initially, and the crossover was post-hoc, it cannot be treated as part of a traditional crossover trial design (AB/BA).
Participant flow in the crossover was not randomized to sequence; hence, there could be an order/learning effect occurring, especially with the unbalanced switch. Figure 1 notes 5 control participants were enrolled into the tSCS group and 2 tSCS participants entered into the control group. This should be highlighted again in the discussion so readers do not overinterpret the results.
Materials and Methods, pg 4, line 115. Please include information about the scoring of UEMS (/50). Here (Section 2.1) and in the appropriate figure captions. Also note scoring for the other items (AIS) in the table for the general population to be able to follow.
Materials and Methods, pg 5, line 149. Please explain your choice of simple computer-generated randomization in this study. Based on your small sample size, mixed design complexity (partial crossover / parallel), and possible imbalance of key baseline characteristics, stratified randomization might have been better with baseline respiratory function considered as a possible factor to balance across groups.
Materials and Methods, pg 5, line 152. Asymmetry of crossover participants could influence results, especially if tSCS has lasting neuromodulatory effects.
You noted a 2-week washout in the manuscript. What was your criteria for setting such a duration, as you noted in the limitations that long-term sustainability of the observed improvements is not known?
You also noted a washout of at least 2 months classified a participant as a new participant. Similar to above, please clarify. Does this mean they were included as part of the 7 crossover participants, or not. It is not clear.
In these two instances (2-wk, 2-month washouts), were baseline measurements of the 2nd intervention (after the washout), similar to initial intervention baseline measurements? This could be a way to say that there was no residual neuromodulatory effects from the first intervention.
Materials and Methods, pg 5, line 165. For simple reproducibility, include whether your biphasic pulse was anodic or cathodic-first, and the charge balance.
Results. Was the baseline respiratory function between the 2 groups (statistically) comparable? I do not see any mention of this.
Final proofread needed for some minor errors in grammar and language used.
Author Response
Reviewer 1
General Comment
We thank the reviewer for the careful evaluation of our work and for highlighting important points that will strengthen the manuscript. Below, we provide detailed responses and indicate the changes we will make in the revised version.
Respiratory failure is a leading cause of death after spinal cord injury. Even small improvements can reduce pneumonia risk, hospitalizations, and mortality; hence, this is a very important field needing more research. Cervical transcutaneous spinal cord stimulation (tSCS) to improve respiratory function is a novel and emerging intervention gaining strong theoretical and early clinical support.
The novelty behind this research study is the application of a dual-site cervical tSCS protocol; comparing tSCS during occupational therapy (OT) with a control group undergoing OT alone. The study tracked changes in MIP, MEP, FVC, after eight 1-hr sessions over 2 weeks. Significant improvements in these respiratory measures were seen in the tSCS group. Other spirometric measures did not show any significant change for either group.
Major
Materials and Methods, pg 6, line 192. The current analysis does not clearly describe how the mixed design (partial crossover and parallel components) was handled. Since the study includes both paired (within-subject) and unpaired (between-group) participant types, it’s important to specify which tests were used for which analyses. Including crossover participants in both within- and between-group analyses can violate assumptions of statistical independence and risk Type 1 errors. For example, consider only using the first phase data from crossover participants in the Mann–Whitney U test.
Response:
We thank the reviewer for highlighting this important point. In the revised manuscript, we have clarified our statistical approach and redid some part of the analysis.
“Normality of the data was assessed using the Shapiro–Wilk test. Variables with a normal distribution (UEMS, AIS Total Motor Score, and AIS Total Sensory Score) were analyzed using a two-way mixed ANOVA, with group (tSCS vs. control) as the between-subject factor and time (pre vs. post) as the within-subject factor. Post-hoc comparisons were performed using paired Student’s t-tests to evaluate pre- versus post-intervention changes within each group.
For variables that were not normally distributed (MIP, MEP, MIP%, MEP%, and spirometric parameters within each group), the Wilcoxon signed-rank test was performed for paired comparisons to evaluate pre- versus post-intervention changes within each group., and 95% confidence intervals were calculated for MIP%, MEP%, and spirometric parameters. Effect sizes (Cohen’s d) were calculated for both neurological and respiratory assessments, with values interpreted as follows: 0.2 = small, 0.5 = medium, and 0.8 = large.
To assess score changes, we calculated the difference between pre- and post-intervention values. The Mann-Whitney U test was then used to compare these score changes between tSCS and control group. Spearman's correlation analysis was performed between demographic and baseline variables—including age, UEMS, time since SCI, and tSCS intensity—and score changes in the tSCS group, including score changes in UEMS, AIS total motor score, MIP, MEP, MIP%. MEP%, and spirometric data. All statistical analyses were conducted using SPSS-16, with the significance level set at p < 0.05.”
While acknowledgment in the limitations section demonstrates awareness, the statistical analysis section does not clearly describe how this crossover data was managed to maintain statistical independence for the between-group comparisons. Please clarify and/or consider redoing the statistical analysis to improve robustness of the data presented.
Response:
We now provide the washout periods for the seven individuals, along with their respiratory function assessment values, which were completely different after washout. A specific explanation was also given to justify considering these individuals as new participants. Individuals in the control group could receive tSCS after their initial treatment following a minimum two-week washout. Additionally, two participants from the tSCS group were later included in the control group after a washout of at least two months and were treated as new participants, in line with literature indicating that tSCS effects plateau within two weeks [29] and show no further gains after one month [22].
We added this part in the methodoloy and in the result section: “Five individuals participated in both conditions, with at least a one-month interval between them (range: 1–3 months), always starting with the control condition. In addition, two individuals entered the control group after completing the tSCS intervention, following a washout period of at least two months (range: 2- 4 months) (Table 1).”
Minor
Title. Considering the partial and unplanned crossover design, I feel that the manuscript title is a little misleading, stating that the study is an RCT with "high crossover rate". Please revise to something along the lines of ‘partial crossover design’ or ‘mixed design’.
The number of participants who crossed over doesn’t constitute as a “high” rate. See my comments for the methods section (pg 4, line 91).
Response:
We agree and have revised the title to reflect a partial crossover design rather than “high crossover rate. “Non-Invasive Cervical Spinal Stimulation and Respiratory Recovery After Spinal Cord Injury. Randomized controlled trial with a partial crossover design.”
Introduction, pg 2, line 80. You introduce one of your prior studies, Kumru et al 2023. To follow this as a foundation, it would be nice to read a description of how this current study is a step up. Please highlight the rationale/novelty of this current study.
Response:
We have expanded the Introduction to highlight novelty this part “Previous studies suggest that tSCS may improve respiratory outcomes in cervical SCI, although evidence remains limited to small cohorts and single case report [27,32]. In one study, tSCS was delivered at cervical (C4) and thoracic (T9) levels combined with inspiratory training [27], while Gad et al. [32] applied tSCS at two cervical levels and T1.
Notably, in our previous study on upper-limb function with tSCS at two cervical levels [29], several participants with cervical SCI subjectively reported improved breathing, suggesting a potential respiratory benefit. Although these impressions were unpublished, they motivated us to include respiratory function in the current study. We hypothesize that cervical tSCS applied at two targeted spinal segments, even without respiratory training, may enhance respiratory function by facilitating activation of respiratory muscles in individuals with cervical SCI. These improvements may be associated with factors such as age, upper extremity or total motor score, time since injury, or tSCS intensity.”
Introduction, pg 3, line 85. Please cite prior studies your group performed with cervical SCI, reporting improvements in breathing. You only noted one, but the sentence reads as plural.
Response:
This was a writing error, which has now been corrected..
Materials and Methods, pg 4, line 91. 7 people is not a large absolute number to call the crossover “high”, especially in the absence of a pre-specified crossover design. 7 out of 19 enrolled is ~37%.
Furthermore, since randomization into TSCS and control groups only occurred initially, and the crossover was post-hoc, it cannot be treated as part of a traditional crossover trial design (AB/BA).
Participant flow in the crossover was not randomized to sequence; hence, there could be an order/learning effect occurring, especially with the unbalanced switch. Figure 1 notes 5 control participants were enrolled into the tSCS group and 2 tSCS participants entered into the control group. This should be highlighted again in the discussion so readers do not overinterpret the results.
Response:
We agree and have revised the text to describe the randomized controlled trial with a partial crossover design, rather than “high.” The asymmetry (5 control → tSCS; 2 tSCS → control) is now clearly reported in both the Methods and Discussion, where we caution readers against overinterpretation due to possible order/learning effects: “Third, potential crossover contamination occurred, as 37% (n=7) of participants switched groups. This unbalanced flow raises the possibility of order or learning effects, and thus the findings should be interpreted with caution.”
Materials and Methods, pg 4, line 115. Please include information about the scoring of UEMS (/50). Here (Section 2.1) and in the appropriate figure captions. Also note scoring for the other items (AIS) in the table for the general population to be able to follow.
Response:
We have added details on UEMS scoring (/50) and clarified AIS scoring in both the Methods section and relevant figure/table legends.
Materials and Methods, pg 5, line 149. Please explain your choice of simple computer-generated randomization in this study. Based on your small sample size, mixed design complexity (partial crossover / parallel), and possible imbalance of key baseline characteristics, stratified randomization might have been better with baseline respiratory function considered as a possible factor to balance across groups.
Response:
We thank the reviewer for this point. Due to the small sample size and exploratory nature of the study, we opted for simple randomization. In this study, participants were randomized to groups using simple randomization, and baseline characteristics, including respiratory function, were carefully documented. While stratified randomization based on baseline respiratory function could have helped to further balance groups, the small sample size limited the feasibility of multiple stratification factors.
We added this information in the Methodology: “This study was a randomized controlled trial with a partial crossover design. Randomization was performed using a simple computer-generated procedure to ensure transparency, reproducibility, and minimization of allocation bias.”
Materials and Methods, pg 5, line 152. Asymmetry of crossover participants could influence results, especially if tSCS has lasting neuromodulatory effects.
You noted a 2-week washout in the manuscript. What was your criteria for setting such a duration, as you noted in the limitations that long-term sustainability of the observed improvements is not known?
Response:
According to published data, improvements tend to stabilize after short-term follow-up: in the study by García-Alén et al. (2023), changes plateaued after 2 weeks of last intervention of tSCS, while in Inanici et al. (2021), no further significant gains were observed after 1 month of last intervention in individuals with chronic SCI.
We added this information in the methodology.
You also noted a washout of at least 2 months classified a participant as a new participant. Similar to above, please clarify. Does this mean they were included as part of the 7 crossover participants, or not. It is not clear.
Response:
Thank you for this point. We have clarified that these 7 individuals were considered new participants. This information has now been added to the Results section to address the comment: “Five individuals participated in both conditions, with at least a one-month interval between them (range: 1–3 months), always starting with the control condition. In addition, two individuals entered the control group after completing the tSCS intervention, following a washout period of at least two months (range: 2- 4 months) (Table 1). For this reason, these individuals were considered new participants.
In these two instances (2-wk, 2-month washouts), were baseline measurements of the 2nd intervention (after the washout), similar to initial intervention baseline measurements? This could be a way to say that there was no residual neuromodulatory effects from the first intervention.
Response:
This clarification has been added to the Methods and Limitations. “Participants in the control group could receive tSCS after their initial treatment following a minimum two-week washout. Two participants from the tSCS group were later included in the control group after a washout of at least two months and were treated as new participants, based on literature indicating that tSCS effects plateau within two weeks and show no further gains after one month [22,29].”
Materials and Methods, pg 5, line 165. For simple reproducibility, include whether your biphasic pulse was anodic or cathodic-first, and the charge balance.
Response:
We have now included details on the biphasic stimulation pulses: “………which generates biphasic, charge-balanced rectangular pulses (anodic-first, 1 ms per phase) with a 10 kHz carrier frequency, delivered at 30 Hz.”
Results. Was the baseline respiratory function between the 2 groups (statistically) comparable? I do not see any mention of this.
Response:
Baseline respiratory function did not differ significantly between the two groups. This information has been added to the Results section.
Final proofread needed for some minor errors in grammar and language used.
Response:
We have carefully proofread the revised manuscript to correct language errors and improve clarity.

Reviewer 2 Report
Comments and Suggestions for Authors
The study investigates whether cervical transcutaneous spinal cord stimulation (tSCS) can improve respiratory function after cervical SCI. It is a randomized controlled trial with 19 participants (15 intervention, 11 control, some with crossover). The intervention group received tSCS at C3–C4 and C6–C7 during occupational therapy. The primary outcomes were maximal inspiratory pressure (MIP), maximal expiratory pressure (MEP), and spirometry. Results showed significant improvements in MIP, MEP, and forced vital capacity (FVC) in the tSCS group, but not in controls.
Strengths: Respiratory dysfunction is the leading cause of morbidity in SCI and a non-invasive intervention has translational value.
It is clinically relevant that respiratory strength improved with tSCS even in the absence of concurrent respiratory training.
Major Concerns:
- The washout period is variable and confusing. It is stated as 2 weeks in some locations and 4 weeks at other places in the text. A few individuals crossed over after 2 months, but they were treated as new participants. It is unclear whether 2 weeks is sufficient to eliminate carryover effects. A subset of participants contributing data to both arms make it difficult to interpret the results and may bias results.
- There is a lack of sham condition. Controls received occupational therapy only, while tSCS group was paired with OT. This means differences between groups could reflect additional therapy time and/or attention rather than stimulation itself. Sham stimulation, i.e., with electrodes placed but no active current, would have strengthened the control group and increase validity.
- The authors sometimes refer to “tDCS” instead of tSCS, which is confusing. This should be corrected for clarity.
- Some improvements in neurological scores occurred in control as well as tSCS group. Both groups received rehabilitative training. So it is unclear to what extent respiratory outcomes could be attributed to tSCS.
- Improvements in MIP/MEP were significant, but changes in spirometry beyond FVC were inconsistent. The clinical meaningfulness of the observed increases should be discussed.
- There is a large range of injury severity (AIS score A-D) and acuity (3 mo-6 mo) within both groups. Both of these factors greatly impact anticipated neurological recovery and are confounding factors that should be included in the discussion.
Minor Concerns:
- Some sections (methods and results tables) are dense and difficult to follow. It could use some better structuring or summarizing to improve readability.
- Effect size reporting is inconsistent across tables.
- Some grammatical issues and awkward phrasing should be fixed (e.g., “SCI individuals” could be replaced with “participants with SCI”).
- Ethics approval is mentioned but no clinical trial registration number is provided.
- The neurologic assessment section utilized upper and lower extremity motor and sensory scoring. The correct scale to reference in that case is the International Standards for Neurological Classification of Spinal Cord Injury (ISNCSCI), not the AIS.
Author Response
General Comment
We thank the reviewer for the careful evaluation of our work and for highlighting important points that will strengthen the manuscript. Below, we provide detailed responses and indicate the changes we will make in the revised version.
The study investigates whether cervical transcutaneous spinal cord stimulation (tSCS) can improve respiratory function after cervical SCI. It is a randomized controlled trial with 19 participants (15 intervention, 11 control, some with crossover). The intervention group received tSCS at C3–C4 and C6–C7 during occupational therapy. The primary outcomes were maximal inspiratory pressure (MIP), maximal expiratory pressure (MEP), and spirometry. Results showed significant improvements in MIP, MEP, and forced vital capacity (FVC) in the tSCS group, but not in controls.
Strengths: Respiratory dysfunction is the leading cause of morbidity in SCI and a non-invasive intervention has translational value.
It is clinically relevant that respiratory strength improved with tSCS even in the absence of concurrent respiratory training.
Major Concerns:
- The washout period is variable and confusing. It is stated as 2 weeks in some locations and 4 weeks at other places in the text. A few individuals crossed over after 2 months, but they were treated as new participants. It is unclear whether 2 weeks is sufficient to eliminate carryover effects. A subset of participants contributing data to both arms make it difficult to interpret the results and may bias results.
Response:
We thank the reviewer for this observation. Although our initial methodology specified a two-week washout, in practice all participants were included only after a minimum one-month washout period. We have clarified that these 7 individuals were considered new participants based on the literatures. This information has now been added to the Results section to address the comment:
Methods: Individuals in the control group could receive tSCS after their initial treatment following a minimum two-week washout. Two participants from the tSCS group were later included in the control group after a washout of at least two months and were treated as new participants, based on literature indicating that tSCS effects plateau within two weeks [29] and show no further gains after one month [22].
Results: “Five individuals participated in both conditions, with at least a one-month interval between them (range: 1–3 months), always starting with the control condition. In addition, two individuals entered the control group after completing the tSCS intervention, following a washout period of at least two months (range: 2- 4 months) (Table 1). For this reason, these individuals were considered new participants.
- There is a lack of sham condition. Controls received occupational therapy only, while tSCS group was paired with OT. This means differences between groups could reflect additional therapy time and/or attention rather than stimulation itself. Sham stimulation, i.e., with electrodes placed but no active current, would have strengthened the control group and increase validity.
Response: We agree with the reviewer that the absence of a sham stimulation condition is a limitation of our study. “ the control group received standard occupational therapy alone, while the tSCS group received tSCS during occupational therapy. Therefore, we cannot fully exclude that differences between groups may partly reflect additional therapy time or attention. Future randomized controlled trials including sham stimulation are needed to strengthen internal validity.”
- The authors sometimes refer to “tDCS” instead of tSCS, which is confusing. This should be corrected for clarity.
Response: We thank the reviewer for pointing this out. All typographical errors have been corrected to consistently refer to “tSCS.” - Some improvements in neurological scores occurred in control as well as tSCS group. Both groups received rehabilitative training. So it is unclear to what extent respiratory outcomes could be attributed to tSCS.
Response: We acknowledge the reviewer’s point. Both groups showed improvements in neurological outcomes due to ongoing rehabilitation, which is expected in this subacute SCI population. However, significant gains in respiratory function were observed only in the tSCS group, despite similar neurological and baseline respiratory function across groups. We have clarified this distinction in the Results and emphasized in the Discussion that respiratory improvements likely reflect a specific effect of targeted cervical tSCS, independent of UEMS and total motor score. - Improvements in MIP/MEP were significant, but changes in spirometry beyond FVC were inconsistent. The clinical meaningfulness of the observed increases should be discussed.
Thank you for raising this quesstion: The duration of stimulation was realtivelty short (8 session during two weeks), which could limit it effectivenes. We have now expanded the Discussion to address the clinical significance of MIP and MEP gains, “While improvements in MIP and MEP were significant in the tSCS group, changes in other spirometric measures beyond FVC were less consistent. Nevertheless, the observed increases in respiratory pressures are clinically meaningful, as they reflect enhanced inspiratory and expiratory muscle strength, which may improve cough efficiency, airway clearance, and reduce the risk of respiratory complications in individuals with cervical SCI. These findings warrant further investigation in longitudinal studies with larger patient cohorts.”
- There is a large range of injury severity (AIS score A–D) and acuity (3–6 months) within both groups. Both of these factors greatly impact anticipated neurological recovery and are confounding factors that should be included in the discussion.
Response: We appreciate this important observation. We performed a correlation analysis with the duration of SCI and found no significant relationship. However, we acknowledge that the heterogeneity in injury severity and time since injury remains a potential limitation that may have influenced the outcomes. This point has now been added to the limitations sectio:” Another limitation was the wide range of injury severity (AIS grades A–D) and time since SCI (3–10 months) within both groups. Although no significant relationship was found between changes in respiratory function scores and time since SCI, both factors can influence the expected neurological recovery and therefore represent potential confounding variables.”.
Minor Concerns
- Some sections (methods and results tables) are dense and difficult to follow. It could use some better structuring or summarizing to improve readability.
Response: We restructured the tables and also highlighted the significant changes in the figure to improve readability.In addition, in response to reviewers’ comments, we included further methodological and results details to address their critiques and ensure completeness.
- Effect size reporting is inconsistent across tables.
Response: We have revised the tables to ensure that effect sizes are reported consistently throughout. For datasets with pre- and post-values, we did not find any inconsistencies in effect size reporting. Could you please specify where the inconsistency is? - Some grammatical issues and awkward phrasing should be fixed (e.g., “SCI individuals” could be replaced with “participants with SCI”).
Response: We thank the reviewer for this suggestion. We have replaced “SCI individuals” with “participants with SCI”. - Ethics approval is mentioned but no clinical trial registration number is provided.
Response: We have now included the trial registration: “This study was registered at ClinicalTrials.gov (Identifier: NCT07140354)” in the article. - The neurologic assessment section utilized upper and lower extremity motor and sensory scoring. The correct scale to reference in that case is the International Standards for Neurological Classification of Spinal Cord Injury (ISNCSCI), not the AIS.
Response: We thank the reviewer for highlighting this. We have corrected the text to reference the ISNCSCI scale appropriately.

Reviewer 3 Report
Comments and Suggestions for Authors
In this manuscript, the authors present the results of a study investigating the impact of a two-week programme of non-invasive electrical stimulation of the cervical spinal cord on respiratory function in individuals with spinal cord injuries of varying severity. The study protocol was carefully planned and executed with clarity. The results are particularly valuable because a subgroup of participants initially in the control group underwent a rehabilitation course in the experimental group after a break. These results will undoubtedly be of interest to researchers specialising in respiratory physiology, motor control physiology and rehabilitation. However, the manuscript contains ambiguities in meaning and several formal shortcomings, and needs to be improved.
The text should be significantly revised.
- The manuscript describes how stimulation improved the participants' respiratory function, but it does not provide information on the degree of respiratory dysfunction in the participants. This information should be included, as should a discussion of the dependence of the changes obtained on the starting point.
- The participants' parameters are presented in such a way that it is unclear which participants were initially in the experimental group and then moved to the control group, or vice versa. This needs to be clarified for these participants, along with the changes that occurred during the break. This information is important for understanding the duration of the stimulation effect on motor and respiratory functions, and it should be discussed.
- Significant improvements in motor function were observed in both the main group, who underwent stimulation, and the control group, who did not. After the course, there were no differences between the groups in terms of motor function indicators. How does this correlate with previous studies showing that cervical stimulation dramatically improves hand function compared to rehabilitation without stimulation [Ref. 29 in this manuscript; Benavides et al. J. Neurosci. 40, 2633–2643 (2020)]? However, the Discussion does not address motor function at all. This is a shortcoming of the article.
- Bipolar pulses were used for stimulation. Consequently, during each pulse, each electrode acts as both an anode and a cathode. When the electrodes are placed at the back of the neck and at the front above the iliac crests, those on the front surface act as cathodes for half of the stimulation time, stimulating the abdominal muscles.
Contraction of the abdominal muscles affects the parameters of external respiration. In study [Minyaeva et al. Hum Physiol 2019, 45, 262–270], a decrease in exhalation duration was observed with non-invasive stimulation of the lumbar spinal cord, and abdominal muscle contraction was cited as a possible cause of this result. While your discussion attributes the stimulation result exclusively to the activation of spinal networks, the possible contribution of the abdominal muscles should also be considered.
- Line 49. Please update references 2 and 3. There are newer studies and reviews on respiratory spinal networks available than those from 1985 and 1994. The referenced studies also focus on the respiratory systems of animals.
- Line 169. What pulse shape was used to determine the thresholds? Were conventional rectangles or bipolar pulses modulated at a kilohertz frequency used for stimulation during the course? Study [Yang et al. Clinical Neurophysiology Practice. S2467981X25000186 (2025)] shows that motor thresholds determined by these different pulses can differ significantly.
- Lines 269, 295, 328 and 329. What does the abbreviation tDCS mean?
- Tables 3 and 5. What are the units of maximal inspiratory and expiratory pressure? What do MIP% and MEP% mean?
- Line 313. What is the percentage relative to? Is it relative to pressure in absolute units or in percentages?
Author Response
Reviewer 3
General Comment
We thank the reviewer for the careful evaluation of our work and for highlighting important points that will strengthen the manuscript. Below, we provide detailed responses and indicate the changes we will make in the revised version.
Comments and Suggestions for Authors
In this manuscript, the authors present the results of a study investigating the impact of a two-week programme of non-invasive electrical stimulation of the cervical spinal cord on respiratory function in individuals with spinal cord injuries of varying severity. The study protocol was carefully planned and executed with clarity. The results are particularly valuable because a subgroup of participants initially in the control group underwent a rehabilitation course in the experimental group after a break. These results will undoubtedly be of interest to researchers specialising in respiratory physiology, motor control physiology and rehabilitation. However, the manuscript contains ambiguities in meaning and several formal shortcomings, and needs to be improved.
The text should be significantly revised.
- The manuscript describes how stimulation improved the participants' respiratory function, but it does not provide information on the degree of respiratory dysfunction in the participants. This information should be included, as should a discussion of the dependence of the changes obtained on the starting point.
Response:
We agree with the reviewer. In the revised manuscript, we will add detailed baseline respiratory data (e.g., forced vital capacity, maximal inspiratory pressure [MIP], maximal expiratory pressure [MEP]) to clarify the degree of dysfunction.
Since baseline respiratory function was similar between groups and improvements occurred only in the tSCS group, we did not include a discussion on whether the observed changes depended on baseline values.
“Baseline values of MIP, MEP, MIP%, MEP%, and spirometric measurements did not differ between the control and tSCS groups (p > 0.05, Mann–Whitney U test).
According to predicted values for MIP%, at baseline in the tSCS group, 3 individuals (20%) had normal inspiratory strength, while 12 (80%) showed pathological inspiratory strength. Following the last session, one additional participant achieved normalization, resulting in 4 individuals (26.7%) with normal MIP%. For MEP%, all 15 individuals with SCI (100%) exhibited pathological expiratory strength (35), with no changes observed after the last session.
In the control group, 2 individuals (20%; 1 female) had normal inspiratory strength (MIP%), while 8 (80%) showed pathological inspiratory strength. For MEP%, all 10 individuals with SCI (100%) exhibited pathological expiratory strength (35). These proportions in MIP% and MEP% remained unchanged following the last session.”
- The participants' parameters are presented in such a way that it is unclear which participants were initially in the experimental group and then moved to the control group, or vice versa. This needs to be clarified for these participants, along with the changes that occurred during the break. This information is important for understanding the duration of the stimulation effect on motor and respiratory functions, and it should be discussed.
Response:
We thank the reviewer for pointing out this ambiguity. We have revised the Methods and Results sections to clearly identify these individuals and we added next information:
“Individuals in the control group could receive tSCS after their initial treatment following a minimum two-week washout. Two participants from the tSCS group were later included in the control group after a washout of at least two months and were treated as new participants, based on literature indicating that tSCS effects plateau within two weeks [29] and show no further gains after one month [22].”
“Five individuals participated in both conditions, with at least a one-month interval between them (range: 1–3 months), always starting with the control condition. In addition, two individuals entered the control group after completing the tSCS intervention, following a washout period of at least two months (range: 2- 4 months) (Table 1). For this reason, these individuals were considered new participants.”
Significant improvements in motor function were observed in both the main group, who underwent stimulation, and the control group, who did not. After the course, there were no differences between the groups in terms of motor function indicators. How does this correlate with previous studies showing that cervical stimulation dramatically improves hand function compared to rehabilitation without stimulation [Ref. 29 in this manuscript; Benavides et al. J. Neurosci. 40, 2633–2643 (2020)]? However, the Discussion does not address motor function at all. This is a shortcoming of the article.
Response:
We agree this is a shortcoming. We will include a dedicated subsection in the Discussion to interpret our motor function results. Specifically, we will compare our findings with prior studies (e.g., Inanici et al. 2021; Garcia-Alen et al., 2023, Moritz et al. 2024) who used repeated tSCS sessions who combined tSCS with occupational therapy and explore potential reasons why our cohort showed improvements in both groups. In this part, we did not include the study by Benavides et al. (2020), which applied one session of 20 minutes of tSCS without concurrent occupational therapy; the primary outcomes were neurophysiological assessments, and upper extremity motor scores were not evaluated.
“In our study, significant improvements in UEMS were observed in both the tSCS and control groups, with no significant differences in change scores. Similarly, García-Alén et al. (29) reported improvements in UEMS and GRASSP, although only GRASSP changes were significantly greater in the tSCS group.
By contrast, three studies in chronic SCI reported gains included improvements in hand force and/or function (15,22,30), motor and sensory abilities (30), and grip strength increases of approximately twofold without stimulation and threefold during stimulation (15) in controlled (22,30)or un control condition (15). The improvements observed in the tSCS group were expected; however, similar gains in the control groups of both our study and that of García-Alén et al. (29) may be explained by the fact that participants were within the first year post-SCI, a period when spontaneous recovery is still possible (47). In contrast, the studies by Inanici and Moritz et al. (22,30) included individuals more than one year post-injury, a stage when recovery is typically limited (47,48).”
- Bipolar pulses were used for stimulation. Consequently, during each pulse, each electrode acts as both an anode and a cathode. When the electrodes are placed at the back of the neck and at the front above the iliac crests, those on the front surface act as cathodes for half of the stimulation time, stimulating the abdominal muscles.
Contraction of the abdominal muscles affects the parameters of external respiration. In study [Minyaeva et al. Hum Physiol 2019, 45, 262–270], a decrease in exhalation duration was observed with non-invasive stimulation of the lumbar spinal cord, and abdominal muscle contraction was cited as a possible cause of this result. While your discussion attributes the stimulation result exclusively to the activation of spinal networks, the possible contribution of the abdominal muscles should also be considered.
Response:
We appreciate this important observation. In the revised Discussion, we will acknowledge the potential contribution of abdominal muscle activation. We will also cite Minyaeva et al. (2019) and discuss how abdominal contractions could shorten exhalation duration and contribute to changes in external respiration, in addition to spinal network modulation.
“On the other hand, bipolar tSCS electrodes over the iliac crests may directly activate expiratory abdominal muscles, potentially shortening exhalation and influencing external respiration (46). Although we did not observe overt abdominal contractions in our study, we cannot exclude a contribution from anode-induced activation at the iliac crests, which may improve respiratory function.”
- Line 49. Please update references 2 and 3. There are newer studies and reviews on respiratory spinal networks available than those from 1985 and 1994. The referenced studies also focus on the respiratory systems of animals.
Response:
We will replace the older references with more recent studies as suggested.
Billing I, Foris JM, Card JP, Yates BJ. Transneuronal tracing of neural pathways controlling an abdominal muscle, rectus abdominis, in the ferret Brain Res 1999 ;820(1-2):31-44. doi: 10.1016/s0006-8993(98)01320-1.
Smith JC, Abdala AP, Koizumi H, Rybak IA, Paton JF. Spatial and functional architecture of the mammalian brain stem respiratory network: A hierarchy of three oscillatory mechanisms. J. Neurophysiol. 2007;98:3370–3387. doi: 10.1152/jn.00985.2007.
- Line 169. What pulse shape was used to determine the thresholds? Were conventional rectangles or bipolar pulses modulated at a kilohertz frequency used for stimulation during the course? Study [Yang et al. Clinical Neurophysiology Practice. S2467981X25000186 (2025)] shows that motor thresholds determined by these different pulses can differ significantly.
Response:
Thank you for noting this. We will specify this in the Methods:” tSCS was delivered using the BioStim-5 transcutaneous electrical stimulator (Cosyma Inc., Moscow, Russia), which generates biphasic, charge-balanced rectangular pulses (anodic-first, 1 ms per phase) with a 10 kHz carrier frequency, delivered at 30 Hz.”
“For single-pulse stimulation, we used biphasic, charge-balanced pulses, anodic-first (positive phase preceding negative) with a 10 kHz carrier frequency.”
- Lines 269, 295, 328 and 329. What does the abbreviation tDCS mean?
Response:
We apologize for this error. The abbreviation should have been tSCS (transcutaneous spinal cord stimulation), and this has now been corrected in the revised manuscript.
- Tables 3 and 5. What are the units of maximal inspiratory and expiratory pressure? What do MIP% and MEP% mean?
Response:
Thank you for pointing this out.
MIP and MEP will be expressed in cmH₂O. MIP% and MEP% represent values expressed as percentages of predicted norms adjusted for age, sex, and body size.
We will clearly explain this in the table legends and Methods.
- Line 313. What is the percentage relative to? Is it relative to pressure in absolute units or in percentages?
Response:
Thank you for pointing this out.
It was MIP% and MEP% represent values expressed as percentages of predicted norms adjusted for age, sex, and body size.We will clarify this explicitly in the text.

Round 2
Reviewer 2 Report
Comments and Suggestions for Authors
Authors have addressed the concerns.
Reviewer 3 Report
Comments and Suggestions for Authors
Dear Authors,
Thank you for revising the manuscript. The manuscript is now clear and relevant, and is presented in a well-structured manner.
Best regards,
Reviewer